# Being Active during the Lockdown: The Recovery Potential of Physical Activity for Well-Being

**DOI:** 10.3390/ijerph18041707

**Published:** 2021-02-10

**Authors:** Clément Ginoux, Sandrine Isoard-Gautheur, Claudia Teran-Escobar, Cyril Forestier, Aïna Chalabaev, Anna Clavel, Philippe Sarrazin

**Affiliations:** 1School of Human Movement & Sport Sciences, University of Grenoble Alpes, SENS, F-38000 Grenoble, France; claudia.teran-escobar@univ-grenoble-alpes.fr (C.T.-E.); cyril.forestier@univ-lemans.fr (C.F.); aina.chalabaev@univ-grenoble-alpes.fr (A.C.); anna.clavel@univ-grenoble-alpes.fr (A.C.); 2School of Political Sciences, University of Grenoble Alpes, PACTE, F-38000 Grenoble, France; 3Laboratoire Motricité, Interactions, Performance, MIP—EA4334, Le Mans Université, 72085 Le Mans, France

**Keywords:** well-being, COVID-19 lockdown, need satisfaction, physical activity, recovery, perceived stress, subjective vitality

## Abstract

To examine the indirect relationships between physical activity, and well-being (i.e., need satisfaction, subjective vitality, and stress) of individuals, through recovery experiences (i.e., detachment from lockdown, relaxation, mastery, and control over leisure time) during the spring 2020 COVID-19 lockdown. Methods. 405 participants answered an online survey including questions on physical activity, recovery experiences, subjective vitality, perceived stress, and basic psychological needs satisfaction. Structural equation modeling tested a full-mediated model in which physical activity predicted recovery experience, which in turn predicted well-being. Results. Physical activity was positively related to a latent variable representing recovery experiences, which in turn was positively related to a latent variable representing well-being. Conclusions. Physical activity carried out regularly during the COVID-19 lockdown positively predicted well-being through recovery experiences. The study results highlight the importance of maintaining or increasing physical activity during periods when recovery experiences and well-being may be threatened.

## 1. Introduction

Since December 2019 and the emergence of the coronavirus 2019 disease (COVID-19), nearly 90 countries have adopted lockdown as a measure to prevent the spread of the virus. At the time of writing (22 November 2020), there have been 57.8 million cases of COVID-19 worldwide, and 1,377,395 deaths recorded [1]. The drastic measure of lockdown was not without psychological consequences. In France, the government announced a national lockdown from 17 March to 11 May. Other countries around the world (e.g., England, Spain, Germany, China, Australia, India, Canada) adopted the same strategy to contain this epidemic. People were authorized to leave their homes for work, medical check-ups, essential purchases, or exercise (for less than one hour per day and less than one kilometer from their home) and carrying a signed certificate detailing the reason for being away from their domicile. The lockdown situation often implies a separation from friends and family, loss of freedom through being confined to restricted space, uncertainty about the evolution of the epidemic, and the modification of work with a possible overlap between private and professional life for those who telework. All these disturbances can have a significant impact on individuals’ well-being, defined as an optimal psychological experience and functioning [2]. Empirical findings have shown that lockdown is related to decreased well-being, such as emotional disturbances [3], depressive symptoms [4], stress [5] and emotional exhaustion [6]. In this context, it is crucial to identify the individual strategies that can help to improve or maintain well-being during periods of lockdown. Many people felt the need for regular physical activity (PA) during the lockdown.

The COVID-19 lockdown has affected the PA habits of many individuals. With the closure of sports facilities, some significantly reduced their PA levels. Conversely, others used the lockdown as an opportunity to initiate PA, both because they had more time to practice, and because it was one of the only reasons for them to go out of their homes during lockdown [7,8,9]. For example, in France, a recent study [7] showed that lockdown affected usual levels of PA for nearly 8 out of 10 French people, where 42% of the French population reported a decrease in their usual level of PA, while 37% increased their usual level of PA. Physical activity has often been described as essential for well-being and crucial for a healthy lifestyle [10]. Indeed, it has been widely demonstrated that PA contributes to higher levels of well-being [11,12,13]. From a theoretical point of view, it is possible to refer to the Conservation of Resources Theory [14], widely used in the literature, to understand how PA can help gain new resources and improve individuals’ well-being. This theory states that resource gain is particularly important in times of resource loss (such as during a lockdown), and that this resource gain could stop the loss spiral. Physical activity may have played a significant protective role in the health and well-being of individuals. The practice of PA and the associated recovery experiences could be a way to prevent the depletion of individuals’ resources by bringing in new resources and participating in a spiral of resource gain. In a context of resource loss such as the COVID19 lockdown, this could be particularly relevant. Practicing PA during the COVID19 lockdown had positive consequences on the well-being of individuals. Some studies have shown a negative relationship between PA and mental health or well-being during lockdown [15,16,17,18]. In their study, Cheval et al. [18] reported a negative relationship between time spent on sedentary activities and mental health or vitality. Contrary to the authors’ hypothesis, the study reports only a low correlation between PA during lockdown and well-being. It is possible that during this period, PA itself is not a direct and automatic predictor of well-being. Of the studies published to date on the relationship between PA and well-being during lockdown related to COVID-19, none have examined the mechanisms explaining how PA can improve the well-being of individuals. Since this relationship could be mediated by other mechanisms, studying these could help to identify a possible sequential process in which PA is indirectly related to better well-being. This could help understand why PA during lockdown can be beneficial for people’s well-being. The recovery experiences are based both on the Effort-Recovery Model [19,20] and the Conservation Of Resources theory [14,21]. The first postulates that employees use their personal resources during efforts they make at work and are only able to recover when they distance themselves from their job and its demands. The second postulates that individuals are intrinsically motivated to protect the resources they have and try to invest them to gain new ones. In light of these two models, PA could be used in a recovery perspective and act as a buffer against negative effects of the lockdown, to replenish used resources. It could also be used for the generation of new resources that may increase well-being regardless of depletion of resources. According to the findings by Sonnentag et al. [22,23], the effects of leisure-time PA on well-being could be mediated through numerous recovery experiences. Further research is needed to explore whether PA and recovery experiences played a role in well-being during this pandemic lockdown. The present study aims to extend the current literature, by examining the mediating role of recovery experiences on the relationship between PA carried out during lockdown and well-being. This study provides multiple contributions to the existing literature. Although some studies have focused on the recovery hypothesis in relation to PA, this study is the first to propose an exploration of this relationship during lockdown. In addition, this study examines the relationships between PA and all recovery experiences identified in the literature. This study also demonstrates that it is possible to consider a general recovery experience subjacent to different recovery experiences, emphasizing that it may be relevant to consider these recovery experiences jointly, rather than separately [23,24]. Finally, unlike most existing studies that adopted a within-person perspective, we adopted a between-person perspective. In contrast to within-person comparisons which provide information on the evolution of the well-being of the same individual over time, between-person comparisons provide a better understanding of why some individuals report higher well-being than others concerning their practice of PA during lockdown.

### 1.1. Physical Activity and Well-Being

Well-being is a subjective and multifaceted experience [2,25,26] and several indicators have been used to infer it. According to Ryan and Deci [2], it could be accurately represented by the satisfaction of needs for autonomy (i.e., to experience a sense of choice and freedom to engage in an activity), competence (i.e., to feel able to effectively carry out challenging tasks), and relatedness (i.e., to develop meaningful relationships with the social environment and acceptance by significant others), and subjective vitality (i.e., a positive and phenomenologically accessible state of having energy available to the self [25]). Cohen et al. [26] argued that (low) perceived stress could also be considered as a well-being indicator (i.e., the psychological strain or distress resulting from exposure to unusual or demanding situations, known as stressors; [27]). Although the beneficial effects of PA on well-being have been demonstrated [12,13], the mechanisms underlying these effects are still poorly understood. Studies have examined the mechanisms by which PA is likely to impact well-being, and some physiological and psychological hypotheses can be used to explain this relationship [28]. Among them, the recovery experiences hypothesis has been well studied and has highlighted the role of numerous recovery experiences on the well-being of individuals [22].

### 1.2. Recovery Experiences as a Mediator of the Physical Activity—Well-Being Relationship

In the work context, studies have demonstrated the existence of several experiences that allow employees to recover more effectively from work demands in their leisure time [22,23]. Recovery experiences refer to “how” people live their leisure activities and what they feel during them. Four types of recovery experiences have been identified by Sonnentag et al. [22]. Psychological detachment refers to the subjective experience of taking psychological distance from a specific context/situation during free time. Relaxation refers to a state of low activation and increased positive affect, allowing for physical and psychological decompression that individuals may experience in their free time. Mastery refers to experiences that allow individuals to acquire new skills or perform meaningful tasks that ultimately increase their self-esteem and confidence. Control over leisure time refers to a person’s ability to decide what, when, and how to do what they want to do in their free time. Research has shown that all these recovery experiences were positively related to well-being and that the occurrence of these recovery experiences during leisure time was related to improved well-being in the following hours/days [23,29,30,31,32,33].

Research has shown that PA is associated with higher psychological detachment, higher levels of relaxation, and greater experience of mastery [23,34,35,36,37]. Moreover, PA could be particularly helpful for the experience of control, as individuals can choose which PA they engage in, and can control various parameters (e.g., duration, intensity, location, whom they exercise with). Research has shown that the beneficial effect of PA on well-being could mainly be explained by the recovery experiences that individuals had during their PA [37,38]. However, PA, recovery experiences and their relationship to well-being have not yet been explored during a situation such as the COVID-19 lockdown. In this situation, it can be assumed that, in this particularly exhausting, context, recovery experiences play a central role in promoting well-being. Thus, this study seeks to address knowledge gaps that may positively benefit an understanding of well-being mechanisms during the COVID-19 pandemic. Understanding the contribution of these key factors may clarify why some individuals are more likely to experience well-being, which will highlight solutions to improve the well-being of individuals during current and future waves of COVID-19 infection and future pandemics. Although previous studies have examined the role of recovery experiences in the relationship between leisure activities (including PA) and well-being, the impact of lockdown on the relationship between these variables is unknown. 

### 1.3. Objectives and Hypothesis

The objective of the present study is to examine whether recovery experiences mediate the relationship between PA during the COVID-19 lockdown and the well-being of individuals. We hypothesized that the relationship between PA during lockdown and well-being is fully mediated by the experience of detachment and relaxation from the COVID-19 situation, and mastery and control over leisure time [23]. Specifically, we assume that PA during lockdown will be related to individuals’ well-being only if it predicts higher recovery experiences [37,38,39]. To ensure this, we will test a model in which PA is expected to predict recovery experience which, in turn, will predict well-being.

## 2. Materials and Methods

### 2.1. Participants and Procedure

This study was conducted following ethical principles specified in the APA (American Psychological Association) Standards. Participants were contacted by networking and social media with a brief text presenting the study protocol and inclusion criteria. The survey was available between 30 March (two weeks after the French government announced the lockdown) and 10 April 2020. Volunteers were invited to copy-paste a URL link through the survey website. First, participants had to read the informed consent (see Appendix A) and to click on the continue button to accept it and access the questionnaire. The survey assessed sociodemographic and environmental variables, well-being, PA, and recovery experiences during leisure time. Participants were told that, for each fully completed questionnaire, the laboratory is committed to donate 0.50 euros to fund a pilot bio-clinical study on biomarkers of COVID-19 aggravation. A total of 405 participants completed the entire online survey. The sample was composed of 279 women (68.8%) with a mean age of 34.06 years (SD = 14.18). Two hundred and thirty-seven participants (58.5%) were still working during the lockdown, of which 168 (41.8%) were teleworking. Three hundred and four participants (75.1%) had an outside space at home (e.g., balcony, terrace, or private garden) and the average housing size was 104.65 m^2^ (SD = 62.3). One hundred and twenty-three participants (33.4%) were parents with children at home, and the average number of children was 0.62 (SD = 1.12).

### 2.2. Measures

#### 2.2.1. Physical Activity

Physical activity before lockdown

The usual level of PA before the lockdown was assessed using the Saltin-Grimby Physical Activity Questionnaire [40]. Participants answered the following question: “Before lockdown, how much did you usually move and exert yourself physically during leisure time? If your activity varied greatly from week to week, try to estimate an average”. Participants had to choose an answer from Level 1 (almost completely inactive) to level 4 (regular hard physical training for competitive sports).

Physical activity during the lockdown

Physical activity during lockdown was assessed with the International PA Questionnaire (IPAQ, [41]), which was adapted to better reflect the circumstances of the COVID-19 lockdown. This questionnaire is more suitable for assessing physical activity levels over the previous seven days than the Saltin-Grimby questionnaire presented above. Participants answered the following questions: “Over the past 7 days, indicate the time spent in minutes for each listed type of physical activity”: Walking outside; running outside; climbing the building/house’s stairs several times; doing muscle strengthening exercises (abs, push-ups, squats) or balance/stretching exercises (tai chi, yoga); cycling, rowing or doing cardio activities at home. These categories were chosen based on a recent opinion article about how to maintain PA levels during COVID-19 [42]. Participants were also asked to add the time spent doing any other physical activities and, in this case, to define these activities. We then classified each activity into moderate-to-vigorous physical activity (MVPA) when it was greater or equal to 3 METs (Metabolic Equivalent of Task) using the compendium of physical activities of Ainsworth et al. [43]. The weekly time spent on MVPA was then calculated and used in the analyses.

#### 2.2.2. Recovery Experiences

We assessed four types of recovery experiences using a French version [44] of the Recovery Experience Questionnaire [22] adapted for the lockdown related to the COVID-19 pandemic. Participants were asked the following question, “During the last week of lockdown, to what extent would you say that the physical activity you did allowed you to (…)”. The 12 items in this questionnaire allow the following to be measured: psychological detachment (4 items, e.g., “(…) distanced myself from the current situation”; α = 0.81), relaxation (4 items, e.g., “(…) to unwind and relax”; α = 0.84), mastery (4 items, e.g., “(…) to learn new things”; α = 0.80), and control (4 items, e.g., “(…) to decide my own schedule”; α = 0.88). The answers are given using a five-point Likert scale (1 = strongly disagree; 5 = strongly agree).

#### 2.2.3. Well-Being

Subjective vitality

Participants completed the French version [45,46] of the Subjective Vitality Scale [25]. This scale began with the stem: “Indicate to what extent each of the following sentences reflects the general feelings you had during the past seven days”. It was composed of five items (e.g., “I felt alive and vital”; α = 0.88) and responses ranged on a seven-point Likert scale (1 = strongly disagree; 7 = strongly agree).

Perceived stress

We used the French version [47] of the four-item shortest version of the Perceived Stress Scale (PSS; [47,48]) to assess general perceived stress during the past seven days. The items measure the frequency with which respondents find their lives unpredictable, uncontrollable, or overwhelming (e.g., “How often have you felt that you could not control the important things in your life?”; α = 0.83) with a 7-point Likert scale (1 = never; 7 = very often).

Basic Psychological Needs Satisfaction

We used the three need satisfaction subscales of the French version [49] of the Basic Psychological Need Satisfaction and Frustration Scale (BPNSFS; [50]). This tool contains 12 items that capture autonomy satisfaction (4 items; e.g., “I feel a sense of choice and freedom in the things I undertake”), competence satisfaction (4 items; e.g., “I feel confident that I can do things well”), and relatedness satisfaction (4 items; e.g., “I feel that the people I care about also care about me”) with a 5-point Likert scale (1 = completely disagree; 5 = completely agree). This scale showed good reliability (α = 0.85).

#### 2.2.4. Demographic and Control Variables

Sociodemographic and environmental information included age, gender, number of children, face-to-face work (0 coded for “no”, 1 coded for “yes”), teleworking (0 coded for “no”, 1 coded for “yes”), housing size, and housing with or without outside space.

#### 2.2.5. Statistical Analyses

The hypotheses were tested using Structural Equation Modeling (SEM) with the Lavaan package ([51]; version 0.6-3) in R [52] (the R script and the dataset for this research can be found in the platform Open Science Framework, at the following link: https://osf.io/6fr5j/). Considering correlations among the four recovery experiences (i.e., 0.53 < r < 0.72, Table 1) and to prevent multicollinearity issues, we decided to model recovery experience as a latent variable whose indicators would be the perceptions of detachment, relaxation, mastery, and control. Considering the correlations between psychological need satisfaction, subjective vitality, and perceived stress (*r*_s_ from −0.45 to 0.45, Table 1), we decided to create a latent variable representing well-being.

Using a two-step method, confirmatory factor analysis was first used to assess the measurement model and was followed by an assessment of the hypothesized model [53]. This approach first establishes the fit of the measurement model by examining the relationship of the observed variables (e.g., recovery experiences and well-being) to their underlying constructs. According to modification indices, covariances between variables were added to improve the model fits [54]. Second, after a satisfactory fit was achieved for the measurement model, we tested the fit of the structural model (i.e., the presumed relationships between the different variables measured). The models’ fit was assessed by examining the comparative fit index (CFI), the Tucker–Lewis-Index (TLI), and the root-mean-square error of approximation (RMSEA). A satisfactory model fit is indicated by a CFI over 0.90 and an RMSEA below 0.05 [55].

We tested an SEM model in which PA is presumed to predict recovery experience which in turn predicts well-being (see Figure 1), supposing an indirect relationship between PA and well-being through recovery experiences. Some authors have pointed out that, from a statistical point of view, there may be an indirect relationship between two variables, through other variables, without there being initially a direct relationship between these two variables [56]. This is particularly the case in the condition of a mediation-only effect. In this model, we added sex (women were coded 0), age, number of children, access to an outside space (no was coded 0), housing size, employment status, and teleworking as control variables for MVPA, recovery experiences, and well-being. The usual level of PA before the lockdown was added as a control variable for MVPA and well-being to control the effect of pre-lockdown behaviors.

Finally, an alternative model in which the direct path between PA and well-being was added was carried out to confirm the presupposed total mediation effect. Fit indices of the hypothesized model and the alternative model were compared. Direct, indirect, and total effects were then computed (Appendix A).

## 3. Results

Descriptive analyses (means and standard deviations) and correlation matrices are presented in Table 1 (the correlation matrix of all variables included in this study is available in Appendix A). The mean of MVPA was 400.25 min per week (SD = 325.02). As MVPA did not have a normal distribution, a square root transformation was applied to approximate a normal curve. Once MVPA was transformed, skewness and kurtosis were examined to check for normality. MVPA was correlated with psychological detachment, relaxation, mastery, control over leisure time, needs satisfaction, subjective vitality, and PA before lockdown. MVPA was not significantly correlated to perceived stress. In addition, the four recovery experiences were significantly and positively correlated to need satisfaction and subjective vitality, and negatively to perceived stress (except for detachment).

First, the measured model was tested. Model fit indices were poor (χ^2^ (13) = 92.401, RMSEA = 0.126, 95% CI (0.102; 0.150), CFI = 0.927, TLI = 0.882). According to modification indices, we added covariances between psychological detachment and relaxation, and between psychological detachment and mastery, to improve the model fits [54]. Thus, the SEM model yielded an acceptable fit to the data, χ^2^ (11) = 41.097, RMSEA = 0.084, 95% CI (0.058; 0.112), CFI = 0.972, TLI = 0.947. In a second step, we added paths between latent variables, directly-observed variables, and control variables (see Figure 1). Then, control variables were added in the model. Finally, non-significant paths were removed from the model, to keep the most parsimonious model, according to the methods described by MacCallum [57]. In Figure 1, only significant paths are presented. This SEM model yielded an acceptable fit to the data χ^2^ (38) = 91.226, RMSEA = 0.060, 95% CI (0.045; 0.076), CFI = 0.956, TLI = 0.940. Results of the structural model revealed that MVPA during lockdown is predicted by PA before the lockdown (β = 0.341, *p* = 0.001), accounting for almost 12% of the variance (R^2^ = 0.116). In addition, weekly time spent on MVPA predicted positively recovery experiences (β = 0.345, *p* = 0.001) and accounted for almost 12% of the variance (R^2^ = 0.116). In turn, well-being was predicted by recovery experiences (β = 0.549, *p* = 0.001), sex (β = −0.217, *p* = 0.001) and age (β = 0.143, *p* = 0.005). These three variables explained 37% of its variance (R^2^ = 0.373). More precisely, these results indicate that, after controlling for sex and age, recovery experiences positively predict well-being. To confirm the presupposed total mediation effect, we carried out an alternative model in which the direct path between PA and well-being were added. Results showed that this path was not significant (β = −0.016, *p* = 0.770) and that the addition of this direct path did not significantly improve the model (χ^2^ (1) = 0.085, *p* = 0.771). The less parsimonious model (i.e., the alternative partially mediated model) was rejected in favor of the fully mediated model that was hypothesized originally. Indirect and total effects are displayed in Table 2.

## 4. Discussion

### 4.1. Main Findings

The objective of the present study was to examine the relationship between PA, recovery experiences, and well-being during the COVID-19 lockdown. Moreover, it investigated whether recovery experiences mediate the relationship between PA and the well-being of individuals. Results showed that recovery experiences felt during PA are positively related to well-being (i.e., psychological need satisfaction, subjective vitality, and stress), while controlling for possible covariates. Besides, our study highlighted an indirect-only mediation effect of recovery experiences on the relationship between PA during the lockdown and well-being. These last results highlight that individuals who engage in more PA and who have more recovery experiences as a result of this PA have higher chances of reporting high levels of well-being. In addition, findings revealed that the usual level of PA before lockdown is moderately positively related to PA during the COVID-19 lockdown, accounting for 12% of its variance.

### 4.2. Theoretical Implications

Findings showed that individuals who were the most active before lockdown were the most active during the COVID-19 lockdown, but the strength of this relationship is moderately weak. The percentage of explained variance highlights that the level of PA during the COVID-19 lockdown did not depend exclusively on previous usual levels of PA. This is in line with the idea that a context change can disrupt existing habits [18,58]. These results highlight that other psychological, sociodemographic, and environmental factors could predict PA during the lockdown, in addition to before-lockdown PA habits. Besides, our results confirmed that weekly time spent on MVPA during lockdown is positively related to recovery. This result is consistent with previous work that had shown positive relationships between PA and recovery experiences [34,35]. Our results are in agreement with the theoretical foundations presupposed by Sonnentag [10] assuming that PA is positively associated with recovery experiences. Moreover, they reinforce these theoretical postulates by emphasizing that this relationship remains significant even in an anxiety-provoking context, such as the COVID19 lockdown.

Further, results revealed that recovery experiences felt during PA can promote well-being. This result is in line with existing results [23,37,38,40,41] showing that experiences during leisure time are associated with well-being. Consistent with the results of some existing studies [29,59], our study highlights that this relationship is not only observable for work-related well-being, but also with general well-being indicators. This result is also in line with the theoretical contributions of Sonnentag [23], arguing that the activities individuals perform and the experiences they have during leisure time are related to their general and work well-being. From a theoretical point of view, these results strengthen the idea that recovery experiences are related to well-being. They also suggest that the theoretical framework of recovery experiences can be extended to the non-work context, and that these recovery experiences could also predict general well-being, and not only work-related well-being. Finally, these results extend the application of the theoretical postulates formulated by Sonnentag [23], emphasizing that recovery experiences are important for the development of the recovery effect of PA on well-being in a lockdown context. This last point indicates that, in order to promote well-being during lockdown, it would be more relevant to look for many recovery experiences (psychologically detaching oneself from stressors, relaxing, learning or mastering new things, deciding when, how, where, and with whom to practice) during one’s PA practice, rather than focusing on the total duration of the PA. In addition, it should be noted that our results have revealed that recovery experiences are positively related to well-being, controlling for sex and age, which are both significantly related to well-being. Indeed, this study shows that women report lower levels of well-being than men and that younger people report lower levels of well-being than older people. Carrying out regular PA during the lockdown and the recovery experiences felt while doing so could therefore be particularly useful and relevant to the well-being of women and younger people. On this point, in her latest review, Sonnentag [23] has underlined that sex and age have not received much attention in the literature, other than as control variables. One possible explanation is that the recovery processes are closely linked to mood regulation and affective dispositions, which could differ over aging, and between men and women. Thus, the effectiveness of specific recovery activities or experiences could be different according to age and sex. Note that our results do not show any significant relationship between the number of children, access to an outside area, house size, face-to-face work, and teleworking on the one hand, and PA or well-being on the other.

Recovery experiences are full mediators of the relationship between PA and well-being. This indirect-only mediation [56] is consistent with theoretical postulates from the research field of recovery, which argue that recovery experiences explain the beneficial effect of recovery activities, such as PA, on individuals’ well-being [22,23]. In addition, our findings are also consistent with the results of previous studies which have shown an indirect-only mediation through recovery experiences [38,39]. Our results enhance these theoretical propositions and provide a better understanding of “how” PA is related to individual well-being. In other words, the practice of PA in which individuals do not experience recovery has limited or no relevance for the well-being of individuals. Conversely, when PA involves recovery experiences, it represents a benefit to the individual’s well-being. Beyond specifying how PA is positively related to well-being, this result reinforces theoretical assumptions and highlights that in an exceptional health context such as the COVID-19 pandemic lockdown, the positive relationship between PA and well-being is dependent on the presence of recovery experiences during PA. This result supports the theoretical postulates that recovery experiences are closer to well-being than recovery activities. Thus, to implement the theory, it would be necessary to examine whether certain PA practice conditions foster the expression of these recovery experiences. This question would be particularly relevant in a context of a lockdown since PA facilities are restricted.

### 4.3. Strengths and Limitations

The present study has two main strengths. First In contrast to previous studies, ours observed the recovery process through three indicators of general well-being: psychological need satisfaction, subjective vitality, and stress. These indicators directly represent both positive and negative indicators of well-being. Then, our study was interested in the recovery experiences felt during PA and their links to well-being during the lockdown. This provided evidence of the importance of these recovery experiences during leisure time for well-being, even in an exceptional health context such as the COVID-19 pandemic lockdown. In addition, in our design, we controlled for PA behavior before the lockdown, allowing us to inspect the effect of PA during the lockdown, considering the usual level of PA. Even if the lockdown could induce changes in PA behavior, we can see that individuals who had higher levels of PA before the lockdown are those who report the higher levels of PA during the lockdown.

Despite these strengths, several limitations deserve to be considered. First, in this study we aimed to consider the whole French adult population, living during the COVID-19 pandemic in Metropolitan France (i.e., under total confinement between March and May 2020). Characteristics of this population included both men and women, at least 18 years of age, representing various socio-professional categories, and continuing, or not, to exercise a professional activity in the workplace or teleworking. The sample we were able to constitute for this study is mainly representative of our reference population, except at some points (see the methods section for a full description), and the self-selected and non-probabilistic nature of the data collection, due to the participants’ recruitment networks, must be mentioned here. Nevertheless, the characteristics (i.e., the distribution of certain descriptive variables) of our sample could prevent us from extending these results to the entire French adult population, particularly for the oldest (i.e., individuals over 60 years), individuals with no outside (garden, balcony, terrace), or individuals with two or more children. Another limitation of this study is its cross-sectional design which does not permit the provision of information about participants’ PA before lockdown. It would have been interesting to examine whether changes in PA levels before the lockdown, compared to after, were related to individuals’ well-being during the lockdown. In addition, it would be necessary to compare individual trajectories of well-being during confinement, distinguishing individuals with a high level of recovery experience from those with low recovery experience. Moreover, it would be relevant to observe the fluctuation of well-being levels and recovery experiences encountered by individuals from day to day during the lockdown, through daily designs. This would indicate whether, on days when individuals have more recovery experiences during PA, they report higher levels of well-being than on days when they engage in PA with fewer recovery experiences. Besides, we used self-reported measures that may include different biases, particularly self-presentation bias [60], self-deception bias [61], and self-ignorance bias [62]. Ideally, measures with experience sampling would have been used to provide more “direct” measures. Nevertheless, in the sanitary situation related to the COVID-19 pandemic, we did not have the latitude and the material resources to implement this type of methodology. Finally, it would be relevant to examine other types of activities performed during lockdown (e.g., creative activities) as it has been shown in the literature that other activities may be able to promote recovery experiences and well-being [23].

## 5. Conclusions

Our study found links between PA, recovery experiences, and well-being during the COVID-19 lockdown. Results demonstrate that recovery experiences fully explain the relationship between PA and well-being. Essentially, maintaining or increasing PA during the lockdown appears as a particularly effective strategy to promote well-being as long as doing so maximizes detachment, relaxation, control, and mastery over free time. From an applied perspective, the results of this study recommend practicing PA offering a high recovery experience to promote well-being. In the event of further lockdowns related to the COVID-19 pandemic, we strongly encourage individuals to engage in distracting and relaxing PA, for which they can choose the duration, the intensity, and the location, and in which they feel competent of progressing.

## Figures and Tables

**Figure 1 ijerph-18-01707-f001:**
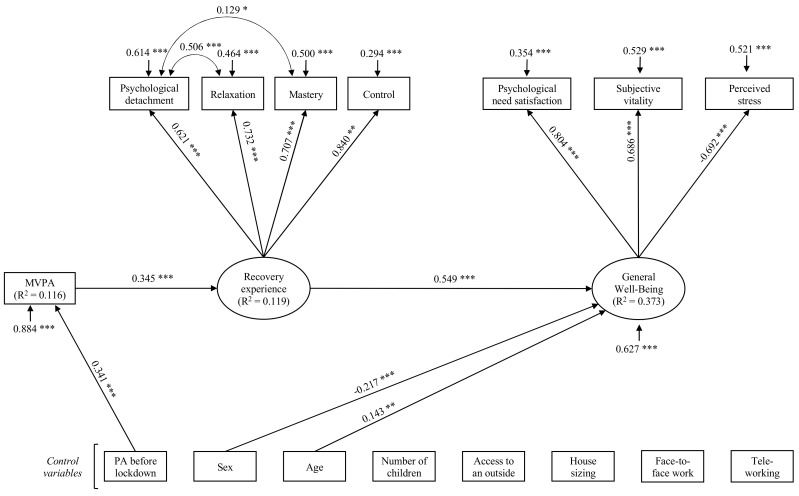
Structural equation modeling of the relations between physical activity, recovery experiences, and general well-being for all participants. Note. Completely standardized maximum likelihood parameter estimates. Solid lines indicate significant paths. Non-significant paths are not displayed. *** *p* < 0.001. ** *p* < 0.01. * *p* < 0.05.

**Table 1 ijerph-18-01707-t001:** Correlation matrix for main variables included in the study.

	M	SD	Observed Range	1	2	3	4	5	6	7	8
1. MVPA	400.52	325.02	0–2130								
2. Usual level of PA	3.02	0.91	1–4	0.34 ***							
3. Detachment	4.61	1.41	1–7	0.24 ***	0.08						
4. Relaxation	5.26	1.28	1–7	0.24 ***	0.12 *	0.72 ***					
5. Mastery	3.90	1.40	1–7	0.25 ***	0.01	0.53 ***	0.53 ***				
6. Control over leisure time	4.93	1.42	1–7	0.30 ***	0.14 **	0.53 ***	0.61 ***	0.59 ***			
7. Need satisfaction	5.30	0.85	1–7	0.16 **	−0.01	0.25 ***	0.31 ***	0.29 ***	0.38 ***		
8. Subjective vitality	4.24	1.27	1–7	0.23 ***	0.03	0.23 ***	0.28 ***	0.36 ***	0.35 ***	0.52 ***	
9. Perceived stress	3.17	1.17	1–7	−0.02	−0.06	−0.03	−0.17 **	−0.13 **	−0.23 ***	−0.57 ***	−0.45 ***

Note. MVPA = Minutes of Moderate to Vigorous Physical Activity per week. PA = Physical Activity. * *p* < 0.05. ** *p* < 0.01. *** *p* < 0.001.

**Table 2 ijerph-18-01707-t002:** Mediation indices for general well-being with recovery experiences as mediator.

Variables	Effect	Estimate	Standardized Estimate	95% CI Lower	95% CI Upper
	General Well-Being				
MVPA	Total	0.016 *	0.177	0.006	0.025
	Indirect	0.017 *	0.193	0.011	0.023
	Direct	−0.001	−0.016	−0.011	0.008

Notes. * 95% confidence interval does not include 0.

## Data Availability

The data presented in this study are openly available in Open Science Framework at https://doi.org/10.17605/osf.io/6fr5j.

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
