# Peer review of "Being Active during the Lockdown: The Recovery Potential of Physical Activity for Well-Being"

_ijerph, 2021, doi:10.3390/ijerph18041707_

Round 1

Reviewer 1 Report

Thank you for the new version of your manuscript. You handled my suggestions well. 

Is it possible that something went wrong regarding formatting/lay-out? 

See line 38 ("separation" doesn't seem to be in place).

See Line 175 (twice the same sentence)

Line 227-258: overlap with previous paragraph

Reviewer 3 Report

In this article, Ginoux et al. report on the indirect positive relationships between physical activity and well-being with recovery experiences during the COVID-19 lockdown as a mediator.  

This survey study is carefully done, and the authors draw suitable conclusions from their results. The limits of the article are well described, particularly the absence of baseline measures and some sampling bias that prevent the generalization of results.

However, the results are in line with other recent studies and may be relevant, especially in the light of the current COVID-19 pandemic.

I still have some minor comments, which I think have to be addressed to improve the article.

-The Introduction section sounds redundant and repetitive (i.e. line 63, 154), which detracts from the paper's clarity. The authors should reduce the generic dissertation to improve the paper readability.

-in the sentence “for each fully completed 168 questionnaire, the laboratory is committed to donate 0.50 euros to fund a pilot bio-clinical study on 169 biomarkers of COVID-19 aggravation” please remove the confounding term “incentive”. It can raise an ethical issue.

-Why did the authors choose different scales to measure physical activity before and during the lockdown? this point should be clarified.

-Line 189 (section Results): the adjective “higher” referred to recovery experiences could be misleading. Please rephrase to make the statement clearer.

-There are several references missing information on volume/issue/page numbers (i.e. number 18, 36). It may be advisable to check the reference for conformity to journal guidelines.

-The self-selected and non-probabilistic nature of the data collection should be mentioned as a further limitation in the Strengths and limitations section.

-I suggest that a native English speaker check this paper language before submitting the revised version.

Author Response

This manuscript is a resubmission of an earlier submission. The following is a list of the peer review reports and author responses from that submission.

Round 1

Reviewer 1 Report

Thank you for your reactions to my comments and an improved version of the manuscript. Although I see that the manuscript has improved, I have a few comments left before I think the manuscript is suitable for publication.

On a general note, I think the introduction can be written in a more concise manner and better structured (especially the introduction section). Each paragraph should begin with the main message. See my comments below for some more detailed suggestions:

Introduction

Line 38-42. Very long sentence. My suggestion would be to make two sentences, i.e., “Seperation from….well-being” & “Well-being can be defined as …”

Line 43: What are “general psychological symptoms”? Maybe better to say that empirical findings indeed show that a lockdown is related to decreased well-being, such as emotional disturbances, depressive symptoms, stress and emotional exhaustion? (this is also more in line with the previous sentences in which you explain that a lock down can have a significant impact on individuals’ well-being).

Line 46-48: This sentence overlaps with previous sentences. Integrate with these previous sentences or remove?

Line 49-53: Physical activity (PA) may have…CoV2 infection”. This information is about the health and well-being effects of PA, while the remainder of the paragraphs is about (changes in) PA levels of people during lockdown. This is little bit confusing. Maybe move this part to the next paragraph in which you write about why PA could have well-being effect, and why these effects may be particularly relevant in times of Covid-19?

Line 53-62: I think this part can be written more concisely. An option could be to start with the main message. In my opinion, this is: For many people, the lockdown affected their usual levels of PA; some people reported a decrease in PA because the venues for PA were closed to the public, while other reported an increase in PA because the lockdown was an opportunity to start new PA, and it was one of the four permitted reasons for leaving home”, and, after that, give some examples + references.

Line 63-64: What is a ‘balanced lifestyle’?

Line 74-75: Here you provide information about PA levels during lockdown. As this paragraph is about well-being effects of PA, I would move this information to the previous paragraph/remove this part (it is little bit distracting from the main message of this paragraph).

Line 85-87: You state that finding mechanisms in the PA-well-being relationship would enable people to cope more efficiently with the lockdown. I would state this a little bit more cautiously/differently, e.g., isn’t it the case that studying mechanisms could help to identify a possible sequential process in which PA is indirectly related to better well-being, and this could help understanding why PA during lockdown would be beneficial for people’s well-being?

Line 93: “recovery hypothesis”. Personally, I am familiar with the recovery hypothesis, but I don’t know if every reader understands this hypothesis. Related to this point, later, you explain that you add to the recovery literature by examining all recovery experiences. I think you have to (shortly) explain the meaning of recovery experiences to the reader earlier in the introduction section. Maybe also integrate recovery and COR theory, and explain that PA can be used to replenish used resources (i.e., recovery perspective, PA may serve as a buffer against negative effects of the lockdown), and help to gain additional (new) resources (resource generation; resources that may increase well-being regardless of depletion of resources)?

Line 94: I doubt whether “Containment context” is correct English?

Line 99-100: You refer to the majority of existing recovery studies. Maybe add some references about recovery (reviews, meta-analyses)?

Line 101-105: The sentences “In contrast to…lockdown”, and “In this article, between-person…individuals” mean the same. Remove one sentence? This is an example how you can write more concisely.

 Line 136-141: I miss the rationale why PA predicts recovery experiences.

Line 140-143: The sentences “Research has shown…their PA”, and “In a usual context…recovery experiences” mean the same. Remove one sentence?

Line 149: The term “ill-being” is new for the reader. Remove?

Discussion

I think if your introduction is written more clearly and better structured, your discussion is also easier to read.

You write that the objective of your study is to examine the relationship between PA, recovery experiences and well-being, but you begin with listing that PA before lockdown is moderately positively related to PA (see Lines 315-317 & Lines 325-331). Personally, I would begin with listing if PA is indeed related to recovery and well-being, as this is the main aim of your study.

I would prefer “theoretical implications” instead of “comparison with other studies”. This requires a little bit re-writing of this section. As a result, as a reader, it is easier to understand what your study adds to the current body of knowledge.

Reviewer 2 Report

I am satisfied with the revisions, and have no further comments.